# Diffusion and Interdiffusion Study at Al- and O-Terminated Al_2_O_3_/AlSi12 Interface Using Molecular Dynamics Simulations

**DOI:** 10.3390/ma16124324

**Published:** 2023-06-12

**Authors:** Masoud Tahani, Eligiusz Postek, Tomasz Sadowski

**Affiliations:** 1Department of Mechanical Engineering, Ferdowsi University of Mashhad, Mashhad 91779-48978, Iran; 2Department of Information and Computational Science, Institute of Fundamental Technological Research, Polish Academy of Sciences, Pawińskiego 5B, 02-106 Warsaw, Poland; epostek@ippt.pan.pl; 3Department of Solid Mechanics, Lublin University of Technology, 20-618 Lublin, Poland

**Keywords:** self-diffusion, interdiffusion, diffusion coefficient, Al_2_O_3_/AlSi12 interface, molecular dynamics

## Abstract

The equivalent characteristics of the materials’ interfaces are known to impact the overall mechanical properties of ceramic–metal composites significantly. One technological method that has been suggested is raising the temperature of the liquid metal to improve the weak wettability of ceramic particles with liquid metals. Therefore, as the first step, it is necessary to produce the diffusion zone at the interface by heating the system and maintaining it at a preset temperature to develop the cohesive zone model of the interface using mode I and mode II fracture tests. This study uses the molecular dynamics method to study the interdiffusion at the interface of α-Al_2_O_3_/AlSi12. The hexagonal crystal structure of aluminum oxide with the Al- and O-terminated interfaces with AlSi12 are considered. A single diffusion couple is used for each system to determine the average main and cross ternary interdiffusion coefficients. In addition, the effect of temperature and the termination type on the interdiffusion coefficients is examined. The results demonstrate that the thickness of the interdiffusion zone is proportional to the annealing temperature and time, and Al- and O-terminated interfaces exhibit similar interdiffusion properties.

## 1. Introduction

Metal matrix composites (MMCs) are increasingly employed in the automotive, aerospace, and biomedical industries owing to their exceptional specific strength, high stiffness, and remarkable wear resistance [1]. These composites commonly employ aluminum, titanium, or magnesium as matrix materials, while alumina, silicon carbide, or boron carbide are often utilized as reinforcing elements [2,3].

Aluminum oxide (Al_2_O_3_) is a versatile and widely used ceramic material with various applications due to its excellent properties and attractive price [4]. Some common uses of aluminum oxide include abrasive material used in grinding and polishing tools, high-temperature environment applications such as furnace linings and refractory materials, electrical insulators, dental and medical applications, and as a filter medium [3,5].

The eutectic aluminum–silicon (AlSi12) alloys, widely used in the transportation industry [6] and have high specific properties and good castability, can replace the pure Al metal matrix. AlSi12 alloy is an aluminum alloy that contains 12 wt.% silicon. It is commonly used in casting applications due to its good fluidity and ability to produce castings with fine details [6]. The high silicon content in the alloy also provides it with excellent thermal properties, making it suitable for use in engine parts and other high-temperature applications [2]. This alloy also has a low density and good corrosion resistance, which makes it useful in the aerospace and automotive industries.

Metal–ceramic composites may exhibit improved wear resistance and strength properties compared to the individual materials and can be used in various high-temperature applications. For example, a composite of Al_2_O_3_ and AlSi12 alloy can be made by various techniques such as powder metallurgy [6,7], hot pressing, squeeze casting [7], or infiltration [1,7,8,9,10]. Interpenetrating phase composites (IPCs) are novel materials with possibly enhanced characteristics compared with traditional composites with discontinuous particles, whiskers, or short fibers [11,12,13,14,15]. The properties and performance of the composite can be tailored by the processing conditions, relative proportions of the two materials, microstructure, proportion of the components, and the interface’s properties.

This study investigates the use of α-Al_2_O_3_ reinforcement in the AlSi12 metal alloy matrix. The Al_2_O_3_/AlSi12 composite has demonstrated very good wear and abrasion resistance [16,17]. Therefore, this composite material has the potential to be used in brake disks in the automotive industry [7]. The mechanical characteristics of the interface constituents and the nature of the interface determine the general mechanical and failure behavior of MMCs [18,19,20]. To this end, the interface attributes in MMCs must be thoroughly investigated. Diffusion causes the interface between phases to exhibit a fuzzy region. Hence, the primary step toward deriving the cohesive zone model of the interface is to investigate the diffusion between the two phases.

Oishi and Kingery [21] first measured oxygen self-diffusion in single and polycrystalline Al_2_O_3_ in 1960. They studied diffusion in temperatures above 1650 °C and observed enhanced diffusion for the polycrystalline specimens. Lagerlof et al. [22] also deduced oxygen self-diffusion coefficients using observations of the shrinking of tiny prismatic dislocation loops in sapphire crystals subjected to prior distortion at a temperature of 1400 °C. The diffusion coefficient was determined, and it was assumed that oxygen lattice diffusion was smaller than aluminum lattice diffusion. Paladino and Kingery [23] determined the self-diffusion coefficient of aluminum in coarse-grain polycrystalline aluminum oxide using aluminium-26 as a tracer in the temperature range of 1670–1905 °C. They found that the diffusivity of aluminum ions is greater than oxygen ions.

Furthermore, Gall et al. [24] measured aluminum self-diffusion in single-crystal α-Al_2_O_3_ using aluminum-26 as a radioactive tracer in the temperature range of 1540–1697 °C. They obtained very different conclusions regarding the diffusion coefficients compared to Paladino and Kingery [23]. A review of the major diffusion processes in α-Al_2_O_3_, including aluminum and oxygen lattice diffusion, oxygen grain boundary diffusion, and pipe diffusion, was presented by Heuer [25]. Knowledge regarding the diffusion of aluminum and oxygen in aluminum oxide was found to be insufficient. Using the density functional theory, Milas et al. [26] investigated the diffusion of Al, O, Pt, Hf, and Y atoms on the α-Al_2_O_3_(0001) surface to study the diffusion mechanisms at the alumina grain boundaries in thermal barrier coatings. They discovered that the Al diffusion is significantly lower than the O diffusion barrier. The literature on the self-diffusion of single crystals and the impurity diffusion of some significant elements in alumina was reviewed by Pelleg [27]. Moreover, they discussed grain boundary diffusion and poly-crystalline alumina diffusion.

Unfortunately, the wettability of ceramic particles with liquid aluminum alloys is often weak. Many technological procedures have been suggested to improve the wetting of ceramic by liquid metal. These include raising the temperature of the metal liquid, pretreatment of ceramic particles or fibers, coating the ceramics, and incorporating some surface-active elements into the matrix. To the authors’ knowledge, no previous investigations have been conducted on the diffusion behavior of the α-Al_2_O_3_/AlSi12 diffusion couple. Therefore, this study aims to explore the self-diffusion and interdiffusion phenomena at the interface by employing the molecular dynamics (MD) method by increasing the system’s temperature to the specified level. The Al- and O-terminated interfaces of α-Al_2_O_3_ with AlSi12 are considered. The influence of annealing temperature, annealing duration, and type of termination at the interface on the diffusion zone and interdiffusion coefficients are studied.

## 2. Modeling Method and Simulation Technique

Diffusion involves the migration of atoms or molecules from a region of higher concentration to a region of lower concentration. Atoms can diffuse across the interface, resulting in the movement of atoms between the two phases. The diffusion rate depends on several factors, including the temperature, chemical composition of the two materials, and the interface between the two phases. At high temperatures, the diffusion rate will be faster, and the atoms will have more energy to move through the material. It is intended to investigate the effect of raising the temperature on the diffusion region and interdiffusion coefficients at the α-Al_2_O_3_/AlSi12 interface.

The molecular dynamics method can study basic processes such as diffusion by using Newton’s second law to calculate the acceleration of atoms by describing atomic interactions through interatomic potentials. In this study, MD simulations are performed utilizing the open-source MD program LAMMPS version 23Jun2022 (large-scale atomic/molecular massively parallel simulator) [28], and the OVITO version 3.8.4 (open visualization tool) software [29] is utilized to visualize the atomic structure’s evolution. The interatomic potential must be precisely quantified because an interatomic potential energy model typically represents atomic interactions. Experimental data or ab initio calculations, such as cohesive energy and elastic modulus, can be used to determine the model parameters. The following section explores the interatomic potentials attributed to aluminum oxide, aluminum, silicon, and the interface.

### 2.1. Potential Functions

In the Al_2_O_3_/AlSi12 system, several atomic interactions are possible and should be taken into account during simulations. The atomic interactions between Al particles in an fcc crystal structure differ significantly from those between aluminum oxide ceramic particles. Metal atoms have electron clouds that determine the strength of their bonds, whereas ionic bonding is the primary factor in ceramics. The interface between metal and ceramic, where the atoms tend to create bonds between two dissimilar structures, introduces additional complexity.

The third-generation charge-optimized many-body potential (COMB3) [30] is a type of interatomic potential that can be used to describe interactions between atoms in aluminum–oxygen systems. The COMB3 potential uses a combination of pair potentials and electron density functions to describe the atomic interactions. The potential is fitted to experimental data and ab initio calculations. It has been shown to reproduce a wide range of properties of aluminum–oxygen systems, including the lattice constant, elastic constants, and the deformation of Al and Al_2_O_3_ under tensile loading. The total energy per atom for the Al-O system, with a charge of *q* at position *r*, in the COMB3 potential can be expressed as [30]:(1)$$U_{{}^{tot}}(r,\;q)=U_{{}^{es}}(q,\;r)+U_{{}^{short}}(q,\;r)+U_{{}^{vdW}}(r)+U_{{}^{corr}}(r)$$
where *U_es_* denotes the energy required to create an atom’s charge, as well as the energies involved in charge–charge interactions, charge–nuclear interactions, and polarizability. Furthermore, *U_short_* is the energy of pairwise attractive and repulsive functions, *U_vdW_* is long-range van der Waals interactions, and *U_corr_* is the correction terms employed to adjust energies associated with specific angles outside the bond order terms.

The Tersoff potential [31], an empirical function composed of two-body terms, is employed for silicon–oxygen interactions. The bonding between atoms *i* and *j* in the many-body Tersoff potential can be expressed as:(2)$$V_{ij}=f_{C}(r_{ij})\left[f_{R}(r_{ij})+b_{ij}f_{A}(r_{ij})\right]$$
where $f_{R}(r_{ij})$, $f_{A}(r_{ij})$, and $f_{C}(r_{ij})$ are repulsive, attractive, and cut-off potential functions, *r_ij_* is the atomic bond length between atom *i* and *j*, and *b_ij_* is a function that adjusts the attractive interaction, respectively.

The ab initio data gathered by Zhao et al. [32] are consistent with the Morse potential, which best represents aluminum–silicon interactions. The Morse potential function is defined as:(3)$$V=D_{0}\left[e^{-2\alpha (r-r_{0})}-2e^{-\alpha (r-r_{0})}\right]$$
where *D*_0_, *α*, *r*, and *r*_0_ represent the well depth of the potential, the width of the potential, the distance between atoms, and the equilibrium bond length, respectively. The Morse potential with parameters *D*_0_ = 0.4824 eV, *α =* 1.322 1/Å, and *r*_0_ = 2.92 Å [26] is employed in this study for aluminum–silicon interactions.

The elastic constants of α-Al_2_O_3_ are determined with previously mentioned potential functions and then compared with the experimental [33], MD simulations [34], and innovative integration of metadynamics and kinetic Monte Carlo simulation techniques ref. [35] in Table 1. The same results for AlSi12 are also presented in this table. The lattice parameters of hexagonal α-Al_2_O_3_ are *a = b =* 4.759 Å, *c =* 12.991 Å, *α = β =* 90^o^, and *γ =* 120^o^, and the lattice constant of fcc Al is 4.0495. AlSi12 single-crystal is formed by substituting 12 wt.% of Al atoms with Si atoms. The linear elastic constants *C_ij_* are obtained at zero temperature by analyzing the stress–strain relation $C_{ij}=\partial \sigma_{ij}/\partial \varepsilon_{ij}$, where $\sigma_{ij}$ and $\varepsilon_{ij}$ are, respectively, the stress and strain components. General decent agreements between the present results and those of other investigators are observed in Table 1. Consequently, the potential functions utilized here demonstrate an accurate simulation of the interactions between atoms.

### 2.2. Molecular Dynamics Model

According to high-resolution transmission electron microscopy, it has been observed that the predominant orientation relationship at the Al_2_O_3_/Al interface is characterized by the parallel alignment of the Al(111) plane and the Al_2_O_3_(0001) basal plane [36]. Pilania et al. [37] also studied coherent and semi-coherent α-Al_2_O_3_(0001)/Al(111) interfaces with a mixed metallic–ionic atomistic model using MD simulations. Therefore, in this study, the lattice orientation alignment $\left.(0001)[2\;\bar{1}\;\bar{1}\;0{]}_{{\alpha -\mathrm{Al}}_{2}O_{3}}\right\|\;(111)[\bar{1}\;\bar{1}\;2{]}_{\mathrm{AlSi}12}$ is taken into account according to the research of other investigators.

The current model comprises a bilayer nanocomposite composed of α-Al_2_O_3_ and AlSi12. The initial α-Al_2_O_3_/AlSi12 interface shown in Figure 1 is considered a single crystal of AlSi12 at the bottom and a single crystal of α-Al_2_O_3_ at the top with an initial gap of 2.0 Å which closely approximates the equilibrium atomic distance at the interface. To examine the impact of alumina terminations on diffusion, two configurations are modeled at the interface: one with Al-termination and another with O-termination. These cases allow a comprehensive exploration to occur of how different terminations affect diffusion behavior. The MD model has a typical size of about 119 × 58 × 184 Å, containing a total of 109,986 atoms.

The geometric arrangement of atoms is optimized through the utilization of the conjugate gradient (CG) energy minimization method. First, the *NVT* canonical ensemble (constant number of particles *N*, volume *V*, and temperature *T*) at a constant temperature of 1200 K is imposed on the sample for 10 ps. Second, the *NPT* ensemble (constant number of particles *N*, pressure *P*, and temperature *T*) at zero pressure and a constant temperature of 1200 K is used for 15 ps to regulate the volume and achieve relaxation in the assembled interface system. Subsequently, the sample is subjected to heating at a heating rate of 10 K/ps until it reaches a preset temperature. Finally, the temperature is held constant at the specified value for a duration of 2.0 ns to analyze interdiffusion while monitoring and recording the atomic movements throughout this period. All processes are conducted using the *NPT* ensemble at zero pressure, employing a time-step of 0.2 fs. The simulations are performed at 1500, 1600, 1800, and 2000 K temperatures. Periodic boundary conditions are implemented for the sample in all three directions.

## 3. Results and Discussion

To study the diffusion properties of the α-Al_2_O_3_/Al interface, the system is heated to a predetermined temperature and maintained there for 2.0 ns. The development of the interface diffusion for the Al-terminated Al_2_O_3_/AlSi12 interface after heating it to 2000 K is illustrated in Figure 2. The initial configuration illustrates the sharp interface between Al_2_O_3_ and AlSi12, considering an initial gap of 2 Å. Furthermore, after maintaining it for 2.0 ns at 2000 K, the system configuration represents the local movement of atoms and the creation of a diffusion zone. The diffusion front is shown in this figure with a dashed line.

### 3.1. Self-Diffusion

The mean square displacements (MSDs) of Al, O, and Si atoms after maintaining the system for a duration of 2.0 ns at different temperatures of 1500, 1600, 1800, and 2000 K for the Al- and O-terminated α-Al_2_O_3_/AlSi12 diffusion couples are depicted in Table 2. This table presents the MSD values for Al atoms in α-Al_2_O_3_, Al atoms in AlSi12, and all Al atoms in the system. As expected, due to the difference in ceramic and metal atomic bonding, the Al atoms in α-Al_2_O_3_ have a significantly lower MSD than the Al atoms in AlSi12. It is also observed that the MSD of O atoms is smaller than the MSD of Al atoms in α-Al_2_O_3_. It is observed that the MSD of Al atoms is larger than the MSD of Si atoms, and the MSD of Si atoms is also larger than the MSD of O atoms.

The coefficients of self-diffusion for each atom type are obtained by analyzing the slope of the MSDs employing Einstein’s relation [38]:(4)$$D_{A}=\underset{t\to \infty }{\lim}\frac{1}{N_{A}}\sum_{i=1}^{N_{A}}\frac{\langle {\left|r_{i}^{A}(t)-r_{i}^{A}(0)\right|}^{2}\rangle }{6t}$$
where *N_A_* represents the total number of atoms of type *A*, $r_{i}^{A}$ denotes the position vector of the *i*th atom belonging to type *A*, and $\langle \cdots \rangle$ signifies the average calculated across all atoms of the same type. The activation energy *Q* and pre-exponential factor *D*_0_ of atoms can be obtained by fitting the self-diffusion coefficients to the Arrhenius equation $D=D_{0}\exp\left(-Q/RT\right)$. The Arrhenius plots of Al, O, and Si atoms for the Al- and O-terminated interfaces are illustrated in Figure 3. Similar to the MSD, Al atoms in Al_2_O_3_ have a significantly smaller self-diffusion coefficient than the Al atoms in AlSi12 because of the differences in atomic bonding between ceramic and metal. Additionally, the self-diffusion coefficient of O atoms is less pronounced than Al atoms in Al_2_O_3_. As can be seen, Al atoms have a higher self-diffusion coefficient than Si atoms, and Si atoms also have a higher self-diffusion coefficient than O atoms. Table 3 also displays the outcomes of the atoms’ activation energies and pre-exponential factors.

### 3.2. Interdiffusion

The interdiffusion flux of an *n-*component system is described by the following Onsager’s formulation [39,40] of Fick’s law:(5)$${\tilde{J}}_{i}=-\sum_{j=1}^{n-1}{\tilde{D}}_{ij}^{n}\frac{\partial C_{j}}{\partial z}$$
where ${\tilde{J}}_{i}$, *C_i_*, and $\partial C_{i}/\partial z$ are the interdiffusion flux, mole fraction, and concentration gradient of component *i*, respectively. Furthermore, ${\tilde{D}}_{ij}^{n}$ is the interdiffusion coefficient. According to Equation (5), the interdiffusion behavior in a ternary system can be described by four independent interdiffusion coefficients: ${\tilde{D}}_{11}^{3},\;{\tilde{D}}_{12}^{3},\;{\tilde{D}}_{21}^{3}$, and ${\tilde{D}}_{22}^{3}$. The Boltzmann–Matano [41,42] method can determine the interdiffusion coefficients.

In the present research, the average interdiffusion coefficients are determined using the approach proposed by Dayananda and Sohn [41]. The average main interdiffusion coefficients (i.e., ${\bar{\tilde{D}}}_{11}^{3}$ and ${\bar{\tilde{D}}}_{22}^{3}$) and cross interdiffusion coefficients (i.e., ${\bar{\tilde{D}}}_{12}^{3}$ and ${\bar{\tilde{D}}}_{21}^{3}$) are evaluated by computing the atomic interdiffusion flux using only the single diffusion couple under study. The concentration curve is fitted using the Gaussian error function for each component. The interested reader will find detailed explanations about the method in Refs. [42,43].

The variations in Al, Si, and O atom concentrations with respect to the *z*-coordinate, which is normal to the interface plane, are shown in Figure 4 for a quantitative analysis of the diffusion process in the Al-terminated α-Al_2_O_3_/Al interface. To obtain the concentration profiles, the diffusion couple is divided into thin slices with a thickness of 2.0 Å along the interface plane. The count of atoms for each type is determined within each slice. Figure 4 shows the initial concentration profiles before diffusion, and the profiles observed after keeping the system at 2000 K for a duration of 2.0 ns. A grey region also depicts the diffusion zone. The variations in atom concentrations for the O-terminated interface, which are not shown here for conciseness, indicate that the diffusion zones in the Al- and O-terminated systems are not significantly different.

Figure 5 illustrates the variations in the interdiffusion flux $\tilde{J}$ and $\tilde{J}(z-z_{0})$ for the Al- and O-terminated α-Al_2_O_3_/AlSi12 diffusion couples after keeping the systems at 2000 K for a duration of 2.0 ns. The position of the Matano plane, denoted by *z*_0_, is also shown in Figure 5 by a vertical dashed line. It is observed that the Matano plane corresponds to the point of highest interdiffusion flux. The independent variables are arbitrarily chosen as the Al and O atoms, while the Si atom is assigned as the dependent variable. The profile variations in the two diffusion couples appear to be very similar. However, it is worth noting that the maximum interdiffusion flux of the Al-terminated interface is slightly higher than that of the O-terminated interface.

Table 4 presents the calculated average values of the main and cross ternary interdiffusion coefficients for the Al- and O-terminated α-Al_2_O_3_/AlSi12 diffusion couples. The diffusion couples are kept at annealing temperatures of 1500, 1600, 1800, and 2000 K for a duration of 2.0 ns. The coefficients are determined using the composition ranges on the lower and upper sides of the Matano plane. It is observed from Table 4 that the main interdiffusion coefficients increase as the annealing temperature increases, as expected. Furthermore, all cross ternary interdiffusion coefficients are significantly smaller, with at least four orders of magnitude lower than the main interdiffusion coefficients. Hence, the cross ternary interdiffusion coefficients do not significantly influence the current ternary systems. Moreover, based on the findings in this table, it is observed that the diffusivity of Si and O atoms shows a slight increase in the Al-terminated system compared to the O-terminated counterpart. However, generally speaking, there is no appreciable distinction between the average interdiffusion coefficients of the Al- and O-terminated systems.

## 4. Conclusions

A molecular dynamics method was employed to investigate atomistic evolutions during the interdiffusion at the α-Al_2_O_3_/AlSi12 interface. The self-diffusion and interdiffusion coefficients were assessed at 1500, 1600, 1800, and 2000 K annealing temperatures for different diffusion couples. Based on the findings of this study, the following conclusions can be made:The self-diffusion coefficient for Al atoms in Al_2_O_3_ is higher compared to O atoms.The average main and cross ternary interdiffusion coefficients were determined for the first time for the Al- and O-terminated Al_2_O_3_/AlSi12 systems utilizing the concentration profiles of atoms during diffusion.The diffusion zone and interdiffusion coefficients increased with the progressive elevation of the annealing temperature and duration.No notable distinction of ternary interdiffusion coefficients was observed between the Al- and O-terminated interfaces.

Future studies may utilize the samples after diffusion and cooling to determine the effective mechanical properties of the Al_2_O_3_/AlSi12 interface through the cohesive zone model and, therefore, the mechanical properties of the MMC.

## Figures and Tables

**Figure 1 materials-16-04324-f001:**
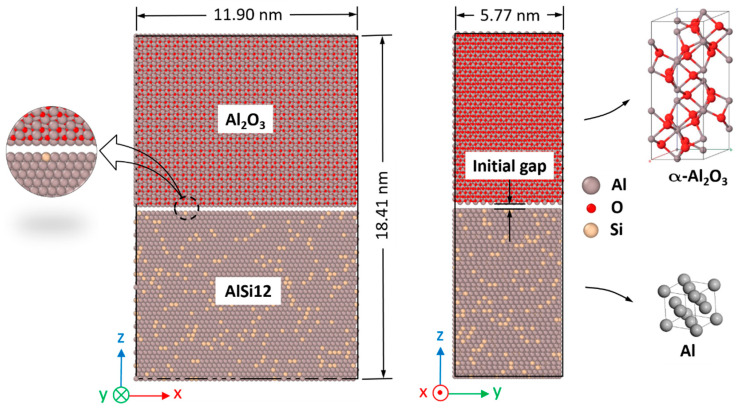
Model of the Al-terminated α-Al_2_O_3_/AlSi12 interface designed for the MD analyses.

**Figure 2 materials-16-04324-f002:**
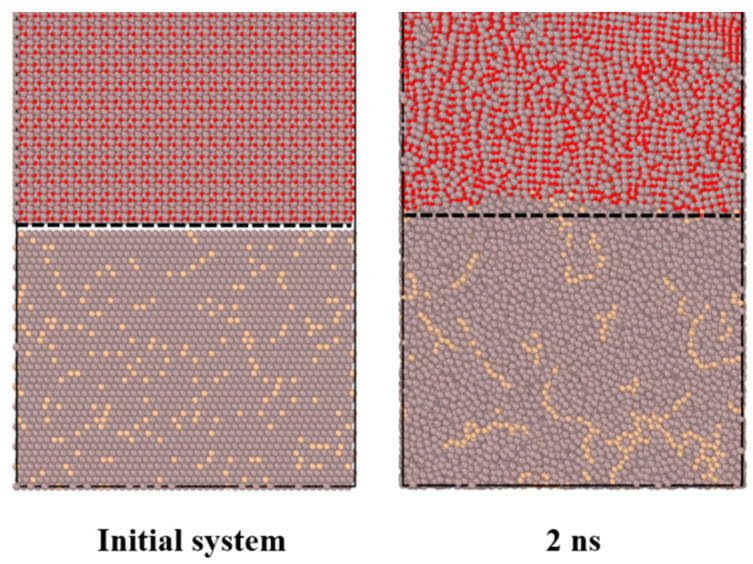
Cross-sections of the atomic configurations of the Al-terminated Al_2_O_3_/AlSi12 interface. The figure shows the initial atomic structure before relaxation, as well as the configuration after the system is held at 2000 K for 2.0 ns. The dashed line indicates the front of the diffusion region.

**Figure 3 materials-16-04324-f003:**
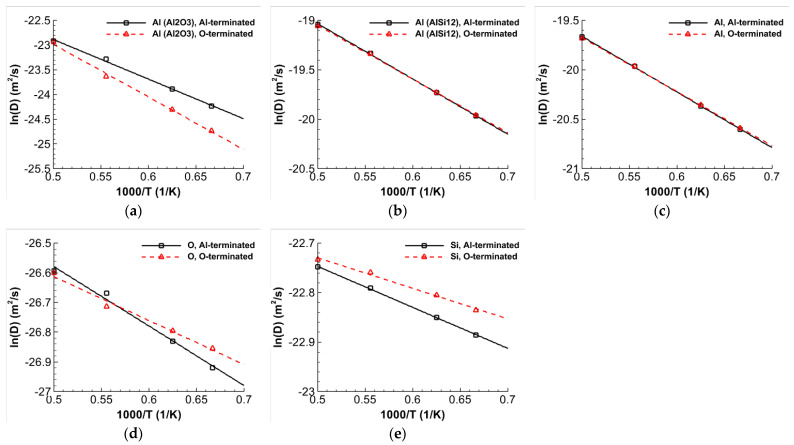
Plots of Arrhenius equation for Al, O, and Si atoms in the Al- and O-terminated α-Al_2_O_3_/AlSi12 interface. (**a**) Al in Al_2_O_3_, (**b**) Al in AlSi12, (**c**) Al, (**d**) O, and (**e**) Si atoms.

**Figure 4 materials-16-04324-f004:**
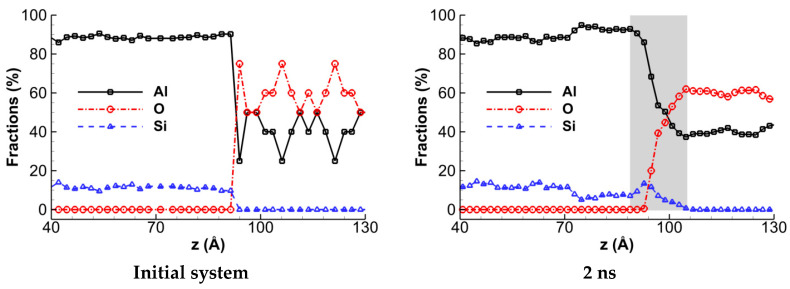
The variations in Al, Si, and O atom concentrations along the *z*-axis during interdiffusion of the Al-terminated α-Al_2_O_3_/AlSi12 interface. The initial system before relaxation and after a 2.0 ns maintenance at 2000 K are illustrated. The diffusion zone is depicted by the gray region.

**Figure 5 materials-16-04324-f005:**
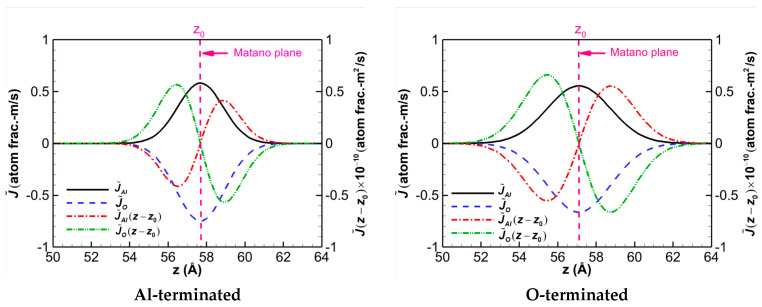
Interdiffusion flux $\tilde{J}$ and $\tilde{J}(z-z_{0})$ for diffusion systems of the Al-terminated and O-terminated α-Al_2_O_3_/AlSi12 heated to 2000 K and kept at that temperature for a duration of 2 ns.

**Table 1 materials-16-04324-t001:** The elastic constants determined through the current MD simulations and their comparison with values reported by other researchers.

Material	Method	*C*_11_ (GPa)	*C*_12_ (GPa)	*C*_13_ (GPa)	*C*_33_ (GPa)	*C*_44_ (GPa)	*C*_66_ (GPa)
α-Al_2_O_3_	Present	510	130	138	518	138	165
	Experiment [33]	497	164	111	498	147	167
	MD [34]	537	180	106	509	130	179
	Monte Carlo simulation [35]	666	269	192	520	158	-
AlSi12	Present	268	134	154	214	108	105

**Table 2 materials-16-04324-t002:** Mean square displacement (nm^2^) of Al, O, and Si atoms at different temperatures for the Al- and O-terminated α-Al_2_O_3_/AlSi12 diffusion couples.

Diffusion Couple	Temperature (K)	Atom				
		Al (Al_2_O_3_)	Al (AlSi12)	Al	O	Si
Al-terminated α-Al_2_O_3_/AlSi12	1500	0.65	24.95	13.46	0.35	2.22
	1600	0.84	32.26	17.45	0.37	2.25
	1800	1.26	47.45	25.51	0.41	2.31
	2000	1.70	63.29	34.01	0.47	2.36
O-terminated α-Al_2_O_3_/AlSi12	1500	0.50	25.11	13.57	0.34	2.34
	1600	0.64	32.26	17.58	0.38	2.36
	1800	1.04	46.98	25.42	0.45	2.40
	2000	1.67	62.34	33.63	0.50	2.46

**Table 3 materials-16-04324-t003:** Arrhenius parameters, *D*_0_ and *Q*, for self-diffusion of Al, O, and Si atoms for Al- and C-terminated α-Al_2_O_3_/AlSi12 diffusion couples.

Atom	Al-Terminated		O-Terminated	
	*Q* (kJ/mol)	*D*_0_ × 10^−9^ (m^2^/s)	*Q* (kJ/mol)	*D*_0_ × 10^−9^ (m^2^/s)
Al (Al_2_O_3_)	66.39	6.32	88.15	21.37
Al (AlSi12)	46.56	89.13	45.22	80.82
Al	46.84	48.15	45.54	44.03
O	16.64	0.0078	12.30	0.0058
Si	6.89	0.200	5.14	0.183

**Table 4 materials-16-04324-t004:** The average interdiffusion coefficients for the ternary systems on either side of the Matano plane. These values are determined after the system is maintained at the preset temperature for a duration of 2 ns.

Diffusion Couple	Temperature (K)	For Composition Range of the Lower Side of Matano Plane ${\bar{\tilde{D}}}_{ij}^{3}\;\times 10^{-11}(m^{2}/s)$				For Composition Range of the Upper Side of Matano Plane ${\bar{\tilde{D}}}_{ij}^{3}\;\times 10^{-11}(m^{2}/s)$			
		${\bar{\tilde{D}}}_{AlAl}^{Si}$	${\bar{\tilde{D}}}_{AlO}^{Si}$	${\bar{\tilde{D}}}_{OAl}^{Si}$	${\bar{\tilde{D}}}_{OO}^{Si}$	${\bar{\tilde{D}}}_{AlAl}^{Si}$	${\bar{\tilde{D}}}_{AlO}^{Si}$	${\bar{\tilde{D}}}_{OAl}^{Si}$	${\bar{\tilde{D}}}_{OO}^{Si}$
Al-terminated α-Al_2_O_3_/AlSi12	1500	0.489	−4.2 × 10^−7^	1.8 × 10^−5^	0.584	0.489	2.2 × 10^−7^	−2.0 × 10^−5^	0.584
	1600	0.518	−6.3 × 10^−7^	8.4 × 10^−6^	0.623	0.518	5.7 × 10^−7^	−9.3 × 10^−6^	0.623
	1800	0.608	−7.9 × 10^−7^	6.1 × 10^−7^	0.845	0.608	−1.2 × 10^−6^	−1.9 × 10^−6^	0.845
	2000	0.753	1.3 × 10^−7^	−6.4 × 10^−6^	1.307	0.753	−2.5 × 10^−7^	2.8 × 10^−6^	1.307
O-terminated α-Al_2_O_3_/AlSi12	1500	0.429	2.1 × 10^−7^	6.3 × 10^−6^	0.489	0.429	2.8 × 10^−7^	4.1 × 10^−6^	0.489
	1600	0.452	−5.4 × 10^−7^	−1.3 × 10^−6^	0.531	0.452	7.6 × 10^−7^	3.8 × 10^−6^	0.531
	1800	0.527	−4.9 × 10^−7^	−6.9 × 10^−6^	0.562	0.527	1.4 × 10^−6^	1.7 × 10^−6^	0.685
	2000	0.696	4.9 × 10^−5^	2.9 × 10^−4^	1.012	0.696	−1.3 × 10^−4^	9.9 × 10^−5^	1.012

## Data Availability

Data available on request due to restrictions, e.g., privacy.

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
