# Peer review of "Diffusion and Interdiffusion Study at Al- and O-Terminated Al2O3/AlSi12 Interface Using Molecular Dynamics Simulations"

_materials, 2023, doi:10.3390/ma16124324_

Round 1
Reviewer 1 Report
The authors present molecular dynamic simulations on the diffusion between Al2O3 and Al-Si12wt% at different temperatures. Theses simulations are important to understand e.g. the behaviour of metal-ceramic composites.
I find the text to be well written and the images to be clear and instructive.
A comment on why the self-diffusion coefficients slightly depend on Al- or O-termination in the case of Al(Al2O3), O and Si but not in the case of Al(AlSi12) would be nice.
The words".is a table." at the end of the description of Table 1 should be removed.
The bibliography is missing. Therfore I can not say weather the cited references are approriate!
Author Response
Dear Respected Reviewer 1,
We sincerely appreciate your valuable and insightful comments on our manuscript. Your comments have proven to be very valuable in improving the overall quality of the paper. Taking into consideration your suggestions, we have made modifications to the manuscript. We have provided detailed responses addressing each of the questions and concerns raised, which are presented below point by point.
Sincerely yours,
Masoud Tahani, Tomasz Sadowski
Reviewer 1
Open Review
(x) I would not like to sign my review report
( ) I would like to sign my review report
Quality of English Language
(x) I am not qualified to assess the quality of English in this paper
( ) English very difficult to understand/incomprehensible
( ) Extensive editing of English language required
( ) Moderate editing of English language required
( ) Minor editing of English language required
( ) English language fine. No issues detected
Yes |
Can be improved |
Must be improved |
Not applicable |
|
Does the introduction provide sufficient background and include all relevant references? |
(x) |
( ) |
( ) |
( ) |
Are all the cited references relevant to the research? |
( ) |
( ) |
( ) |
(x) |
Is the research design appropriate? |
(x) |
( ) |
( ) |
( ) |
Are the methods adequately described? |
(x) |
( ) |
( ) |
( ) |
Are the results clearly presented? |
(x) |
( ) |
( ) |
( ) |
Are the conclusions supported by the results? |
(x) |
( ) |
( ) |
( ) |
Comments and Suggestions for Authors
The authors present molecular dynamic simulations on the diffusion between Al2O3 and Al-Si12wt% at different temperatures. Theses simulations are important to understand e.g. the behaviour of metal-ceramic composites.
I find the text to be well written and the images to be clear and instructive.
Response: We appreciate the positive impression of the esteemed reviewer and have done our best to carefully address all the concerns raised by the respected reviewer.
A comment on why the self-diffusion coefficients slightly depend on Al- or O-termination in the case of Al(Al2O3), O and Si but not in the case of Al(AlSi12) would be nice.
Response: It is important to note that the lattice structure of Al2O3 can initiate with either O atoms or Al atoms. Consequently, the interface between Al2O3 and AlSi12 can only be Al-terminated or O-terminated. However, in the case of AlSi12, we only have Al atoms and a small fraction of Si atoms at the interface due to the replacement of 12 wt.% of Al atoms by Si atoms. As a result, it is not appropriate to refer to the interface as Si-terminated.
Furthermore, the self-diffusion coefficients of the Al-terminated and O-terminated interfaces exhibit slight differences due to differences in the atomic interactions between Al (in Al2O3) and Al (in AlSi12), as compared to the atomic interactions between O (in Al2O3) and Al (in AlSi12).
If we have not fully grasped the viewpoint expressed by the esteemed reviewer, we kindly request a more comprehensive explanation to enable us to thoroughly analyze the raised issue.
The words".is a table." at the end of the description of Table 1 should be removed.
Response: Thank you very much for accurately reading the manuscript. “.is a table” was removed in the revised manuscript.
The bibliography is missing. Therfore I can not say weather the cited references are approriate!
Response: According to the concern raised by the respected reviewer, we decided to search the literature again for other references that may be missed, and now we are sure that all relevant studies are cited.

Reviewer 2 Report
Authors used molecular dynamics simulation to investigate the diffusion and interdiffusion of α-Al2O3/AlSi12 with different temperature. A single diffusion couple used for each system to study the average main and cross-interdiffusion coefficients. It is found that the thickness of the interdiffusion zone increases with the increasing in annealing temperature and time, and Al- and O-terminated interfaces exhibit similar interdiffusion properties. However, the simulation results don't have enough justification, and thus fail to support authors' conclusion. Based on the quality of the paper and the computational tasks performed, I am sorry to inform the authors that I don't recommend this manuscript to be published on MATERIALS now. My concerns and the suggestions are given below.
1) The title of this manuscript is not appropriate. The title is too general, whereas Authors only studied the diffusion and interdiffusion properties of Al, Si, O atoms of Al2O3 and AlSi12. It seems that a new title which contains or resembles the studied material surface, is more appropriate.
2) Please add the corresponding abbreviation and full name in the article. For example:
Line 100 "LAMPS" and "OVITO"
Line 171 "conjugate gradient"
Line 206 "Mean square displacement"
3) The equilibrium of the system after molecular dynamics simulation needed to be provided into the article. And potential energy and total energy of the Al and α-Al2O3 can be calculated by the analyze software. In addition, the interdiffusion properties of α-Al2O3/AlSi12 at 1500 K, 1800 K and 2000 K have been studied by LAMMPS software, the article can also study the diffusion properties at more temperature such as 1000 K, 1200 K and 1600 K.
4) After continuous heating and simulated equilibrium, the interaction energy between Al2O3 and AlSi12 have a certain change. With the temperature increase, Al2O3 and AlSi12 will gradually approach each other. The van der waals, electrostatic potential energy and total energy should increase gradually. The discussion on the aspects needs to be further strengthened.
5) Some contents listed without reference.
Line 26 "Aluminum oxide (Al2O3) is a versatile and widely used ceramic material with various 26 applications due to its excellent properties and attractive price."
Line 50 "The overall mechanical and failure properties of MMCs depend on interface constituents’ mechanical properties and the interface’s nature."
6) The grammar and vocabulary of English in the text need to be changed. In addition, some common contracts for specific expressions and quantities have been ignored. For example:
Line 63 and line 66 "oC" must be "°C".
Line 172 and line 173 "NVT" must be "NVT", "NPT" must be "NPT"
Line 226 "are the interdiffusion coefficients." must be "is the interdiffusion coefficients."
Line 270 "with increasing the annealing temperature" must be "with increasing of the annealing temperature"
7) In line 217, "because of the differences in atomic bonding between ceramic and metal." Except the diffusion coefficient of Al in Al2O3 and AlSi12, the number of bond formed between Al and O atoms, Al and Si atoms, and the statistics of generated products after heating and simulation can also provide the conclusions. Please adding the corresponding simulation calculations into the article.
8) The annotation in the bottom left corner of and line of Figure 5 are not clear. Please change the thickness and size of the line and upload a new image.
9) There are some long sentences throughout the manuscript that need to rewritten. For example:
Line 261 "The two diffusion couples appear to have very similar profile variations; however, the maximum interdiffusion flux of the Al-terminated interface is slightly higher than that of the O-terminated one."
10) What is the reason for choosing 0.2 fs time step? Usually considered in simulation 1 fs or 2 fs.
11) The motivation for selecting of these ceramic materials has not been determined, clearly. Authors should mention them important. Because, there are many other ceramic materials which can be suitable for this study.
By considering these points, I believe that a major revision is required for this manuscript before any decision for its acceptance or publication.
Authors used molecular dynamics simulation to investigate the diffusion and interdiffusion of α-Al2O3/AlSi12 with different temperature. A single diffusion couple used for each system to study the average main and cross-interdiffusion coefficients. It is found that the thickness of the interdiffusion zone increases with the increasing in annealing temperature and time, and Al- and O-terminated interfaces exhibit similar interdiffusion properties. However, the simulation results don't have enough justification, and thus fail to support authors' conclusion. Based on the quality of the paper and the computational tasks performed, I am sorry to inform the authors that I don't recommend this manuscript to be published on MATERIALS now. My concerns and the suggestions are given below.
1) The title of this manuscript is not appropriate. The title is too general, whereas Authors only studied the diffusion and interdiffusion properties of Al, Si, O atoms of Al2O3 and AlSi12. It seems that a new title which contains or resembles the studied material surface, is more appropriate.
2) Please add the corresponding abbreviation and full name in the article. For example:
Line 100 "LAMPS" and "OVITO"
Line 171 "conjugate gradient"
Line 206 "Mean square displacement"
3) The equilibrium of the system after molecular dynamics simulation needed to be provided into the article. And potential energy and total energy of the Al and α-Al2O3 can be calculated by the analyze software. In addition, the interdiffusion properties of α-Al2O3/AlSi12 at 1500 K, 1800 K and 2000 K have been studied by LAMMPS software, the article can also study the diffusion properties at more temperature such as 1000 K, 1200 K and 1600 K.
4) After continuous heating and simulated equilibrium, the interaction energy between Al2O3 and AlSi12 have a certain change. With the temperature increase, Al2O3 and AlSi12 will gradually approach each other. The van der waals, electrostatic potential energy and total energy should increase gradually. The discussion on the aspects needs to be further strengthened.
5) Some contents listed without reference.
Line 26 "Aluminum oxide (Al2O3) is a versatile and widely used ceramic material with various 26 applications due to its excellent properties and attractive price."
Line 50 "The overall mechanical and failure properties of MMCs depend on interface constituents’ mechanical properties and the interface’s nature."
6) The grammar and vocabulary of English in the text need to be changed. In addition, some common contracts for specific expressions and quantities have been ignored. For example:
Line 63 and line 66 "oC" must be "°C".
Line 172 and line 173 "NVT" must be "NVT", "NPT" must be "NPT"
Line 226 "are the interdiffusion coefficients." must be "is the interdiffusion coefficients."
Line 270 "with increasing the annealing temperature" must be "with increasing of the annealing temperature"
7) In line 217, "because of the differences in atomic bonding between ceramic and metal." Except the diffusion coefficient of Al in Al2O3 and AlSi12, the number of bond formed between Al and O atoms, Al and Si atoms, and the statistics of generated products after heating and simulation can also provide the conclusions. Please adding the corresponding simulation calculations into the article.
8) The annotation in the bottom left corner of and line of Figure 5 are not clear. Please change the thickness and size of the line and upload a new image.
9) There are some long sentences throughout the manuscript that need to rewritten. For example:
Line 261 "The two diffusion couples appear to have very similar profile variations; however, the maximum interdiffusion flux of the Al-terminated interface is slightly higher than that of the O-terminated one."
10) What is the reason for choosing 0.2 fs time step? Usually considered in simulation 1 fs or 2 fs.
11) The motivation for selecting of these ceramic materials has not been determined, clearly. Authors should mention them important. Because, there are many other ceramic materials which can be suitable for this study.
By considering these points, I believe that a major revision is required for this manuscript before any decision for its acceptance or publication.
Author Response
Dear Respected Reviewer 2,
We sincerely appreciate your valuable and insightful comments on our manuscript. Your comments have proven to be very valuable in improving the overall quality of the paper. Taking into consideration your suggestions, we have made modifications to the manuscript. The revised version highlights new or modified sections in BLUE for ease of identification. Additionally, we have provided detailed responses addressing each of the questions and concerns raised, which are presented below point by point.
Sincerely yours,
Masoud Tahani and Tomasz Sadowski
Reviewer 2
Open Review
( ) I would not like to sign my review report
(x) I would like to sign my review report
Quality of English Language
( ) I am not qualified to assess the quality of English in this paper
( ) English very difficult to understand/incomprehensible
( ) Extensive editing of English language required
( ) Moderate editing of English language required
(x) Minor editing of English language required
( ) English language fine. No issues detected
Yes |
Can be improved |
Must be improved |
Not applicable |
|
Does the introduction provide sufficient background and include all relevant references? |
( ) |
(x) |
( ) |
( ) |
Are all the cited references relevant to the research? |
(x) |
( ) |
( ) |
( ) |
Is the research design appropriate? |
(x) |
( ) |
( ) |
( ) |
Are the methods adequately described? |
(x) |
( ) |
( ) |
( ) |
Are the results clearly presented? |
( ) |
(x) |
( ) |
( ) |
Are the conclusions supported by the results? |
( ) |
(x) |
( ) |
( ) |
Comments and Suggestions for Authors
Authors used molecular dynamics simulation to investigate the diffusion and interdiffusion of α-Al2O3/AlSi12 with different temperature. A single diffusion couple used for each system to study the average main and cross-interdiffusion coefficients. It is found that the thickness of the interdiffusion zone increases with the increasing in annealing temperature and time, and Al- and O-terminated interfaces exhibit similar interdiffusion properties. However, the simulation results don't have enough justification, and thus fail to support authors' conclusion. Based on the quality of the paper and the computational tasks performed, I am sorry to inform the authors that I don't recommend this manuscript to be published on MATERIALS now. My concerns and the suggestions are given below.
Response: We would like to express our sincere gratitude to the esteemed reviewer for taking the time to thoroughly examine our manuscript and provide us with invaluable comments that have greatly contributed to enhancing the quality of the final version. Even though we have done our best to modify the manuscript based on your comments, we welcome any further comments if the respected reviewer wishes to raise them.
1) The title of this manuscript is not appropriate. The title is too general, whereas Authors only studied the diffusion and interdiffusion properties of Al, Si, O atoms of Al2O3 and AlSi12. It seems that a new title which contains or resembles the studied material surface, is more appropriate.
Response: According to the suggestion of the respected reviewer, we decided to change the title of the revised manuscript to “Diffusion and Interdiffusion Study at Al- and O-terminated Al2O3/AlSi12 Interface Using Molecular Dynamics Simulations”. Please inform us if you have any additional recommendations for a suitable title.
2) Please add the corresponding abbreviation and full name in the article. For example:
Line 100 "LAMPS" and "OVITO"
Line 171 "conjugate gradient"
Line 206 "Mean square displacement"
Response: The abbreviations were added to the revised manuscript.
3) The equilibrium of the system after molecular dynamics simulation needed to be provided into the article. And potential energy and total energy of the Al and α-Al2O3 can be calculated by the analyze software. In addition, the interdiffusion properties of α-Al2O3/AlSi12 at 1500 K, 1800 K and 2000 K have been studied by LAMMPS software, the article can also study the diffusion properties at more temperature such as 1000 K, 1200 K and 1600 K.
Response: The diffusion properties were determined by allowing the system to reach equilibrium at the desired temperature for a duration of 2.0 ns. In Figure 2, both the initial configuration (before minimization) and the system after equilibration for 2.0 ns at 2000 K are depicted. In the interest of conciseness, the manuscript does not include the equilibrium configurations of the system at other temperatures, as they exhibit a similar overall appearance to Figure 2.
It is important to note that initially, our study focused on a lower temperature range of 1000 to 1500 K. However, the numerical results revealed that interdiffusion is negligible at temperatures below 1200 K. Consequently, we concluded that higher temperatures are required for interdiffusion to occur in pristine diffusion couples (without any defects such as vacancy defects). Therefore, we made the decision to select an annealing temperature range of 1500 to 2000 K. Based on this explanation; we did not include the results for 1000 and 1200 K in the revised manuscript. However, as you suggested, despite the extensive simulation time, we included the results for 1600 K in the revised version.
In your subsequent comment, we elaborated on the changes observed in the potential energy of the system.
4) After continuous heating and simulated equilibrium, the interaction energy between Al2O3 and AlSi12 have a certain change. With the temperature increase, Al2O3 and AlSi12 will gradually approach each other. The van der waals, electrostatic potential energy and total energy should increase gradually. The discussion on the aspects needs to be further strengthened.
Response: According to the comment raised by the respected reviewer, we computed the potential energy of an Al-terminated Al2O3/AlSi12 diffusion system. It is worth mentioning that first the geometric configuration was optimized using the conjugate gradient minimization algorithm. The potential energies at this stage are as follows:
Next, the system was heated from 1200 K to 2000 K, afterwards, the system was maintained at 2000 K for 2.0 ns. The variations of the total potential energy and potential energy of the constituents are presented in the following figure. It can be seen that the potential energy of both constituents generally increased with increasing temperature. However, by maintaining the system at 2000 K for 2.0 ns, the potential energies reduce because the system reaches its equilibrium. It is also observed that the potential energy of Al2O3 is higher compared to that of AlSi12. Furthermore, it should be noted that, in this system, the total number of atoms in Al2O3 and AlSi12 is 73500 and 36645, respectively.
(a)
(b) |
(c) |
- Total potential energy, (b) potential energy of Al2O3, and (c) potential energy of AlSi12 for the Al-terminated Al2O3/AlSi12 diffusion system during heating from 1200 K to 2000 K and maintaining the system at 2000 K for 2.0 ns.
5) Some contents listed without reference.
Line 26 "Aluminum oxide (Al2O3) is a versatile and widely used ceramic material with various applications due to its excellent properties and attractive price."
Line 50 "The overall mechanical and failure properties of MMCs depend on interface constituents’ mechanical properties and the interface’s nature."
Response: In response to the comment provided by the respected reviewer, the references [4] (line 35) and [18-20] (line 62) were added to the revised version of the manuscript for the contents stated above.
6) The grammar and vocabulary of English in the text need to be changed. In addition, some common contracts for specific expressions and quantities have been ignored. For example:
Line 63 and line 66 "oC" must be "°C".
Line 172 and line 173 "NVT" must be "NVT", "NPT" must be "NPT"
Line 226 "are the interdiffusion coefficients." must be "is the interdiffusion coefficients."
Line 270 "with increasing the annealing temperature" must be "with increasing of the annealing temperature"
Response: The corrections were made to the revised version. However, about your third comment here we should mention that we have four interdiffusion coefficients. To this end, it seems that the sentence “Furthermore, ’s are the interdiffusion coefficients.” is correct. Additionally, to ensure the quality of the language used, we revised the whole manuscript carefully to avoid language errors.
7) In line 217, "because of the differences in atomic bonding between ceramic and metal." Except the diffusion coefficient of Al in Al2O3 and AlSi12, the number of bond formed between Al and O atoms, Al and Si atoms, and the statistics of generated products after heating and simulation can also provide the conclusions. Please adding the corresponding simulation calculations into the article.
Response: It is noteworthy that our objective was to emphasize the difference in atomic bonds in Al2O3 and AlSi12, leading us to draw a reasonable conclusion that the MSD of Al atoms in Al2O3 is significantly lower than the MSD of Al atoms in AlSi12 (as illustrated in Table 2).
As per your inquiry regarding the number of bonds, we examined the bond count between atoms in Al2O3 and AlSi12. Our findings revealed that the initial system before heating exhibited 492,800 bonds in Al2O3 and 213,043 bonds in AlSi12. However, upon heating the system to 2000 K, the bond count decreased to 324,791 in Al2O3 and 142,365 in AlSi12. This reduction in bond count suggests that many bonds were broken at this elevated temperature.
In response to your subsequent question regarding the generated products after heating the system, it is worth mentioning that we have previously conducted a separate study on the SiC/Al diffusion couple. In that study, we performed X-ray diffraction (XRD) analysis in the diffused region; however, we were unable to identify specific products based on the XRD pattern analysis. We guess that a more extended equilibration period may be necessary to observe discernible products in the diffused region. The respected reviewer is aware of the computational cost limitations we faced during the MD analysis, particularly in the context of the presented study where we had to employ small time steps of 0.2 fs. Therefore, the time duration in MD simulations is typically constrained to a few nanoseconds.
8) The annotation in the bottom left corner of and line of Figure 5 are not clear. Please change the thickness and size of the line and upload a new image.
Response: Thank you very much for your comment. We increased the font size of the legends and enhanced the thickness of the lines in Figure 5 of the revised manuscript.
9) There are some long sentences throughout the manuscript that need to rewritten. For example:
Line 261 "The two diffusion couples appear to have very similar profile variations; however, the maximum interdiffusion flux of the Al-terminated interface is slightly higher than that of the O-terminated one."
Response: We sincerely appreciate your comment, which has greatly contributed to improving the readability of the revised manuscript. Following your suggestion, we carefully reviewed the entire document and addressed any errors and lengthy sentences. Furthermore, we have revised the sentence you mentioned, and it now appears as follows in the updated version (page 9, lines 285-288).
“The profile variations of the two diffusion couples appear to be very similar. However, it is worth noting that the maximum interdiffusion flux of the Al-terminated interface is slightly higher than that of the O-terminated interface.”
10) What is the reason for choosing 0.2 fs time step? Usually considered in simulation 1 fs or 2 fs.
Response: It is noteworthy that during the initial stages of the simulations, we analyzed the model using a time step of 1 fs. However, we encountered instability issues during the heating period and maintaining the system at elevated temperatures. That is, Lammps was unable to effectively control the temperature at the desired level with using 1 fs time steps. After referring to the Lammps example manual and reviewing other papers that utilized the COMB3 potential (e.g., see Ref. [R1]), we discovered that this potential necessitates much smaller time steps for stability. Despite the resulting increase in total simulation time, we were compelled to utilize this small time step to ensure the stability of our numerical simulations.
11) The motivation for selecting of these ceramic materials has not been determined, clearly. Authors should mention them important. Because, there are many other ceramic materials which can be suitable for this study.
Response: The respected reviewer is aware that aluminum matrix composites belong to the category of metal matrix composites that have undergone extensive research. This is primarily attributed to their remarkable mechanical and thermal properties, along with their high durability, low density, and comparatively affordable raw materials. The Al2O3/AlSi12 composite has demonstrated very good wear and abrasion resistance [R2,R3]. Because of that, this metal matrix composite has the potential to be used in brake disks in the automotive industry [R4]. The driving force behind our study was the distinctive application of this light metal matrix composite, as well as its potential for utilization in various wear resistance scenarios.
To elaborate on the primary rationale behind our choice of this particular metal matrix composite as the subject of our study, the following explanations have been included on page 2, lines 58-60:
“The Al2O3/AlSi12 composite has demonstrated very good wear and abrasion resistance [16,17]. Therefore, this composite material has the potential to be used in brake disks in the automotive industry [7].”
References:
R1. Choudhary, K.; Liang, T.; Chernatynskiy, A.; Phillpot, S.R.; Sinnott, S.B. Charge optimized many-body (COMB) potential for Al2O3 materials, interfaces, and nanostructures. J. Phys. Condens. Matter 2015, 27, 305004, doi:10.1088/0953-8984/27/30/305004.
R2. Dolata, A.J. Tribological Properties of AlSi12-Al2O3 Interpenetrating Composite Layers in Comparison with Unreinforced Matrix Alloy. Materials 2017, 10, 1045.
R3. Tomiczek, B.; Kremzer, M.; Sroka, M.; Dziekońska, M. Abrasive Wear of AlSi12-Al2O3 Composite Materials Manufactured by Pressure Infiltration. Archives of Metallurgy and Materials 2016, doi:10.1515/amm-2016-0207.
R4. Maj, J.; Basista, M.; WÄ™glewski, W.; Bochenek, K.; Strojny-NÄ™dza, A.; Naplocha, K.; Panzner, T.; Tatarková, M.; Fiori, F. Effect of microstructure on mechanical properties and residual stresses in interpenetrating aluminum-alumina composites fabricated by squeeze casting. Mater. Sci. Eng.: A 2018, 715, 154-162, doi:https://doi.org/10.1016/j.msea.2017.12.091.

Reviewer 3 Report
This manuscript contains serious work. It is also well-written. With my comments, I attempt to help the further improvement of it.
2. The symbol of the unit of temperature has been used incorrectly in many places, for example, lines 61, 68, 71. The correct symbol is ℃.
3. In line 104 they write: "using Newton’s second law to describe atomic interactions".
This is not correct. Newton's second law gives the acceleration (i.e. the effect of the sum of the forces)
only if the forces are already given. The formulas of the forces themselves are independent of Newton's second law.
4. Line 105: I think adding some more information about the software, the used time integration methods, the running times etc, would increase the value of the paper.
5. Abbreviations such as NVT and NPT should be resolved.
6. "the sample is heated to a preset temperature at a heating rate of 10 K/ps." As far as I understand, it means that the heating process is longer for larger final temperatures, so the total simulated time is also longer.
The problem is that it influences the conclusion:
"the main interdiffusion coefficients increase with increasing the annealing temperature",
since it is hard to distinguish the effect of larger T and longer simulation.
7. "In Table 1, for AlSi12, the authors did not show any other results than their own, unlike they did for Al2O3. Why?
8. "The periodic boundary conditions are applied in all three directions of the sample."
In the x and y directions, it is OK and natural, but in the z-direction, it is strange a little: it can mean that on the top of the Al2O3, there is AlSi12 without a gap...
9. In Table 2 and later: The Al atoms are either in Al2O3 or AlSi12. So what does Al without the material mean? Do the data represent some average of the Al2O3 and AlSi12 results? Also, the O and Si atoms leave their original material and they diffuse into the other side. Did the authors calculate their diffusion properties there, e.g. the O atoms in AlSi12?
10. In line 279 a new sentence should start: "... one. However, generally speaking ..."
The quality of the English language usage is quite good.
Author Response
Dear Respected Reviewer 3,
We sincerely appreciate your valuable and insightful comments on our manuscript. Your comments have proven to be very valuable in improving the overall quality of the paper. Taking into consideration your suggestions, we have made modifications to the manuscript. The revised version highlights new or modified sections in GREEN for ease of identification. Additionally, we have provided detailed responses addressing each of the questions and concerns raised, which are presented below point by point.
Sincerely yours,
Masoud Tahani, Tomasz Sadowski
Reviewer 3
Open Review
( ) I would not like to sign my review report
(x) I would like to sign my review report
Quality of English Language
( ) I am not qualified to assess the quality of English in this paper
( ) English very difficult to understand/incomprehensible
( ) Extensive editing of English language required
( ) Moderate editing of English language required
(x) Minor editing of English language required
( ) English language fine. No issues detected
Yes |
Can be improved |
Must be improved |
Not applicable |
|
Does the introduction provide sufficient background and include all relevant references? |
(x) |
( ) |
( ) |
( ) |
Are all the cited references relevant to the research? |
(x) |
( ) |
( ) |
( ) |
Is the research design appropriate? |
(x) |
( ) |
( ) |
( ) |
Are the methods adequately described? |
( ) |
(x) |
( ) |
( ) |
Are the results clearly presented? |
( ) |
(x) |
( ) |
( ) |
Are the conclusions supported by the results? |
( ) |
(x) |
( ) |
( ) |
Comments and Suggestions for Authors
This manuscript contains serious work. It is also well-written. With my comments, I attempt to help the further improvement of it.
Response: We appreciate the positive impression of the esteemed reviewer and have done our best to carefully address all the concerns raised by the respected reviewer to improve the quality of the final version of the manuscript.
- The affiliation of the third author is not complete, the department/institute is missing.
Response: We sincerely appreciate your comment. In the revised version, we have ensured that the affiliation of all authors is fully included.
The symbol of the unit of temperature has been used incorrectly in many places, for example, lines 61, 68, 71. The correct symbol is ℃.
Response: Thank you for your thorough review of our manuscript. We made the necessary corrections to ensure that the unit of temperature throughout the manuscript is correct. Furthermore, we carefully examined the entire manuscript to eliminate any other typographical errors.
In line 104 they write: "using Newton’s second law to describe atomic interactions".
This is not correct. Newton's second law gives the acceleration (i.e. the effect of the sum of the forces) only if the forces are already given. The formulas of the forces themselves are independent of Newton's second law.
Response: Thank you once again for your genuine and constructive email, which has helped address the issues in the article. Newton's second law alone does not directly describe the forces acting on atoms within the molecular dynamics method. In molecular dynamics simulations, forces between atoms are typically calculated using interatomic potentials. Newton's second law is then employed to calculate the resulting acceleration of the atoms based on these forces. Therefore, we modified the mentioned sentence in the revised version:“The molecular dynamics method can study basic processes like diffusion by using Newton’s second law to calculate the acceleration of atoms by describing atomic interactions through interatomic potentials.”
4. Line 105: I think adding some more information about the software, the used time integration methods, the running times etc, would increase the value of the paper.
Response: In the manuscript, we have provided details about the number of atoms and the integration methods employed for the equilibration and heating processes (Page 5, lines 187-201). However, we did not specifically discuss the running times, as they can vary significantly based on factors such as CPU configuration, the number of nodes, and the number of CPUs per node in high-performance computing systems. A typical computational time for a high-performance computing system comprising 40 nodes and 48 CPUs per node is approximately 39 hours.
Abbreviations such as NVT and NPT should be resolved.
Response: As suggested by the respected reviewer, the explanations for abbreviations of the NVT and NPT ensembles were added to the revised manuscript (page 5, lines 190-193) as follows:
“Firstly, the NVT canonical ensemble (constant number of particles N, volume V, and temperature T) at a constant temperature of 1200 K is imposed on the sample for 10 ps. Secondly, the NPT ensemble (constant number of particles N, pressure P, and temperature T) at a zero pressure …”.
"the sample is heated to a preset temperature at a heating rate of 10 K/ps." As far as I understand, it means that the heating process is longer for larger final temperatures, so the total simulated time is also longer.
The problem is that it influences the conclusion:
"the main interdiffusion coefficients increase with increasing the annealing temperature",
since it is hard to distinguish the effect of larger T and longer simulation.
Response: We sincerely appreciate your meticulous review of the manuscript. You are right in your observations. To minimize errors when comparing our results, we employed a high initial temperature of 1200 K and a rapid heating rate of 10 K/ps. The heating process for the samples, from the initial temperature to 1500, 1600, 1800, and 2000K, took approximately 0.03 ns, 0.04 ns, 0.06 ns, and 0.08 ns, respectively. Our numerical findings indicated that the diffusion coefficients are not very sensitive to the annealing time. The most significant discrepancy in heating time occurs between the samples at 1500 K and 2000 K, with a difference of 0.05 ns. We believe that this discrepancy is negligible and can be disregarded.
"In Table 1, for AlSi12, the authors did not show any other results than their own, unlike they did for Al2O3. Why?
Response: We had a keen interest in comparing the elastic constants presented for AlSi12 with other references. However, despite conducting extensive literature searches, we were unable to find any suitable results for comparison. Therefore, we believe that our results for elastic constants of AlSi12 are original.
"The periodic boundary conditions are applied in all three directions of the sample."
In the x and y directions, it is OK and natural, but in the z-direction, it is strange a little: it can mean that on the top of the Al2O3, there is AlSi12 without a gap...
Response: We greatly appreciate your comment. In many studies, the diffusion couples have been analyzed using the molecular dynamics method with periodic boundary conditions in the z-direction (e.g., see Refs. [R1-R4]). However, to investigate the impact of shrink-wrapped boundary conditions in the z-direction, we examined this boundary condition by incorporating two thin fixed layers at the top and bottom of the system. Our findings indicate that for larger-sized models along the z-direction, like ours, there is no significant difference in the diffusion results.
We used periodic boundary conditions in the z-direction as it aligns with the approach commonly employed in published papers on diffusion studies. However, if the respected reviewer still believes that non-periodic boundary conditions in the z-direction should be used, we could change the text of the manuscript accordingly.
References:
R1. Zhu, Y.; Liao, G.; Shi, T.; Tang, Z.; Li, M. Interdiffusion cross crystal-amorphous interface: An atomistic simulation. Acta Materialia 2016, 112, 378-389, doi:https://doi.org/10.1016/j.actamat.2016.04.032
R2. Luo, M.; Liang, L.; Lang, L.; Xiao, S.; Hu, W.; Deng, H. Molecular dynamics simulations of the characteristics of Mo/Ti interfaces. Computational Materials Science 2018, 141, 293-301, doi:https://doi.org/10.1016/j.commatsci.2017.09.039.
R3. Xu, R.-G.; Falk, M.L.; Weihs, T.P. Interdiffusion of Ni-Al multilayers: A continuum and molecular dynamics study. Journal of Applied Physics 2013, 114, doi:10.1063/1.4826527.
R4. Horbach, J.; Das, S.K.; Griesche, A.; Macht, M.P.; Frohberg, G.; Meyer, A. Self-diffusion and interdiffusion in Al80Ni20 melts: Simulation and experiment. Physical Review B 2007, 75, 174304, doi:10.1103/PhysRevB.75.174304.
- In Table 2 and later: The Al atoms are either in Al2O3 or AlSi12. So what does Al without the material mean? Do the data represent some average of the Al2O3 and AlSi12 results? Also, the O and Si atoms leave their original material and they diffuse into the other side. Did the authors calculate their diffusion properties there, e.g. the O atoms in AlSi12?
Response: The esteemed reviewer is aware that the behaviour of Al atoms in Al2O3 differs from that in AlSi12 due to the distinct atomic bonding present in these two materials. The atomic interactions between Al particles in an fcc crystal structure differ significantly from those between aluminium oxide ceramic particles. Metal atoms have electron clouds that determine the strength of their bonds, whereas ionic bonding is the primary factor in alumina.
To address this distinction, we employed the practice of using the name of the respective material in parentheses to specify which Al atoms belong to which material. Furthermore, when the name of the material was not specified, the reference to Al atoms represents the average diffusion or interdiffusion behaviour of all Al atoms within the system.
In order to provide clear definitions for Al atoms in the manuscript, an additional sentence was included on page 6, lines 220 and 222, as follows:
“This table presents the MSD values for Al atoms in a-Al2O3, Al atoms in AlSi12, and all Al atoms in the system.”
Figure 4 presents the concentration profiles of all three atom types, demonstrating the movement of atoms across the interface following diffusion. Tables 3 and 4 provide the self-diffusion and interdiffusion data for all atom types, including O and Si atoms. It is worth mentioning that all diffusion properties were obtained by using the concentration profiles of the ternary diffusion system.
- In line 279 a new sentence should start: "... one. However, generally speaking ..."
Response: We have incorporated your suggestion in the revised version to split the long sentences into smaller ones.
